# Comet Assay Profiling of FLASH-Induced Damage: Mechanistic Insights into the Effects of FLASH Irradiation

**DOI:** 10.3390/ijms24087195

**Published:** 2023-04-13

**Authors:** Christian R. Cooper, Donald J. L. Jones, George D. D. Jones, Kristoffer Petersson

**Affiliations:** 1Leicester Cancer Research Centre, University of Leicester, Robert Kilpatrick Clinical Sciences Building, Leicester Royal Infirmary, Leicester LE2 7LX, UK; donald.jones@leicester.ac.uk (D.J.L.J.); gdj2@leicester.ac.uk (G.D.D.J.); 2MRC Oxford Institute for Radiation Oncology, University of Oxford, Old Road Campus Research Building, Oxford OX3 7DQ, UK; 3Department of Haematology, Oncology and Radiation Physics, Skåne University Hospital Lund University, 221 85 Lund, Sweden

**Keywords:** comet assay, DNA damage, electrons, humans, oxygen, radiation, FLASH

## Abstract

Numerous studies have demonstrated the normal tissue-sparing effects of ultra-high dose rate ‘FLASH’ irradiation in vivo, with an associated reduction in damage burden being reported in vitro. Towards this, two key radiochemical mechanisms have been proposed: radical–radical recombination (RRR) and transient oxygen depletion (TOD), with both being proposed to lead to reduced levels of induced damage. Previously, we reported that FLASH induces lower levels of DNA strand break damage in whole-blood peripheral blood lymphocytes (WB-PBL) ex vivo, but our study failed to distinguish the mechanism(s) involved. A potential outcome of RRR is the formation of crosslink damage (particularly, if any organic radicals recombine), whilst a possible outcome of TOD is a more anoxic profile of induced damage resulting from FLASH. Therefore, the aim of the current study was to profile FLASH-induced damage via the Comet assay, assessing any DNA crosslink formation as a putative marker of RRR and/or anoxic DNA damage formation as an indicative marker of TOD, to determine the extent to which either mechanism contributes to the “FLASH effect”. Following FLASH irradiation, we see no evidence of any crosslink formation; however, FLASH irradiation induces a more anoxic profile of induced damage, supporting the TOD mechanism. Furthermore, treatment of WB-PBLs pre-irradiation with BSO abrogates the reduced strand break damage burden mediated by FLASH exposures. In summary, we do not see any experimental evidence to support the RRR mechanism contributing to the reduced damage burden induced by FLASH. However, the observation of a greater anoxic profile of damage following FLASH irradiation, together with the BSO abrogation of the reduced strand break damage burden mediated by FLASH, lends further support to TOD being a driver of the reduced damage burden plus a change in the damage profile mediated by FLASH.

## 1. Introduction

Ultra-high dose rate FLASH radiotherapy (FLASH-RT) is a developing technology set to potentially transform the radiation treatment of cancer [1,2,3]. Compared with conventional dose rate radiotherapy (CONV-RT), which typically delivers dose rates of ca. 0.1 Gy s^−1^ on the timescale of minutes, FLASH-RT uses dose rates several orders of magnitude higher (≥30–100 Gy s^−1^), typically delivered on the millisecond timescale. The very short time of FLASH-RT delivery could help address the issue of organ/tumour motion (by reducing both geographical miss and normal tissue exposure) [4,5], but more importantly, numerous in vivo studies have shown FLASH-RT to lead to the greater sparing of normal tissues, whilst not seemingly compromising tumour control [6,7,8,9,10]. Consequently, FLASH-RT seems set to deliver superior therapeutic ratios compared to CONV-RT. This “FLASH effect” has been observed in numerous in vivo studies [4,6,7,8,9,10,11,12,13], including studies of mice, mini-pigs, zebrafish embryos, and other animals. Recently, one human clinical trial of proton FLASH was completed to assess its feasibility as an alternative treatment for painful bone metastasis [14]; similarly, a dose escalation trial for patients with skin metastasis from melanoma utilizing electron beam FLASH is currently underway [15,16]. However, whilst there is mounting evidence supporting the advantage of FLASH-RT over CONV-RT, the causative underpinning mechanism(s) remain to be elucidated.

The high dose rates of FLASH exposures have been proposed as responsible for an early divergence of radiochemical events that could distinguish FLASH-RT from lower dose rate CONV-RT [17]. At FLASH dose rates, the initial concentration of induced free radicals is expected to be higher than that of conventional exposures, due to the contracted timescale of FLASH delivery. This could alter the corresponding mechanisms of the FLASH- and CONV-induced radiochemical reactions, and consequently the cellular response to the respective irradiations. To date, there have been two key radiation-chemical mechanisms proposed to explain the FLASH sparing of normal tissue [18,19,20]: the radical–radical recombination (RRR) mechanism and the transient oxygen depletion (TOD) mechanism.

The RRR mechanism proposes that the greater ‘instantaneous’ concentration of radicals induced by FLASH exposures leads to higher RRR, ultimately reducing the number of damaging radicals remaining to damage normal tissue [17,21]. Alternatively, the TOD hypothesis proposes that the greater initial concentration of FLASH-induced radicals causes a depletion of local oxygen that occurs on a timescale much faster than tissue/cell re-oxygenation kinetics, leading to a transient local hypoxia, better enabling the chemical repair (notably, thiol-mediated chemical repair) of damaging radiation-induced free radicals [3,22,23,24,25]. Whichever mechanism occurs/prevails, an expected common outcome, resulting from either RRR and/or TOD, is that FLASH irradiation should yield lower levels of induced damage compared to CONV irradiation.

Recently, we reported that FLASH irradiation at low oxygen tension does induce lower levels of DNA damage in whole-blood peripheral blood lymphocytes (WB-PBL) ex vivo, an effect modulated by oxygen tension, dose, and dose rate [26], supporting that an oxygen-related mechanism contributes to the tissue-sparing effect of FLASH irradiation [27]. Nevertheless, this study fails to distinguish between RRR and TOD as being the mechanism(s) responsible for the observed FLASH-mediated reduction in damage burden [26].

A possible outcome of the RRR mechanism is the formation of crosslink damage (**R–R**), notably via organic radical (**R•**) recombination (Reaction (1)) [21].
**R• + R• → R–R**(1)

On the other hand, a possible outcome of TOD, and the consequential transient hypoxia, is a more anoxic profile of the resulting FLASH-induced damage.

Therefore, the objectives of the current study are to profile FLASH-induced damage using, firstly, a modified version of the alkaline Comet assay that allows for the measurement of DNA crosslink formation [28,29], with the detection of crosslinks being a likely indicator of the RRR mechanism occurring, and secondly, to use a further modified version of the alkaline Comet assay to enable the profiling of induced base vs. strand break damage [30], with a relative reduction in the level of induced strand breaks being a possible marker of anoxic DNA damage formation [31,32], indicating TOD. Additionally, with thiol-mediated chemical repair proposed to contribute to the reduced damage burden mediated by FLASH [26], we investigated the buthionine sulfoxamine (BSO) pre-treatment of WB-PBLs to assess what effect lowering cellular thiol levels [33,34] has on the noted reduced damage burden induced by FLASH [26]. A greater relative effect of BSO pre-treatment in abrogating the FLASH-mediated reduced damage burden would indicate that FLASH exposures induce hypoxia, supporting TOD as a mechanism contributing to the FLASH effect.

The ultimate aim of this investigation is to determine the extent to which RRR and/or TOD contribute to the FLASH-induced reduction in strand break damage burden witnessed previously [26].

## 2. Results

### 2.1. Comet Assay Assessment of FLASH-Induced DNA Inter-Strand Crosslinks as a Marker of Radical–Radical Recombination

A variant of the alkaline Comet assay was used to assess FLASH-induced DNA inter-strand crosslink formation as a marker of RRR (Figure 1A). The Comet response (recorded as ‘% Tail DNA’) following 20 Gy CONV irradiation at 21% O_2_ (column 3) was approximately double the response following 10 Gy CONV irradiation at 21% O_2_ (column 4). As expected, there was no significant difference in the Comet response noted between the single dose of 20 Gy CONV irradiation at 21% O_2_ (column 3) and two sequential 10 Gy CONV irradiations at 21% O_2_ irradiations (column 5), indicating the dose-dependent proportionality of the Comet response and the iso-damaging effect of the two sequential 10 Gy CONV irradiations (column 5) vs. a single 20 Gy irradiation (column 3).

As previously noted [26], 20 Gy CONV irradiation at the reduced oxygen tension of 0.5% O_2_ (column 6) led to a significantly (*p* < 0.0001) lower level of induced DNA damage compared to the corresponding irradiation under normoxia (column 3). Furthermore, this was further significantly reduced following 20 Gy FLASH irradiation at 0.5% O_2_ (column 8) (*p* = 0.02), supporting our previous observation of FLASH irradiations inducing lower levels of DNA damage at oxygen tensions of ≤ 1% O_2_ [26].

However, the re-irradiation of both 20 Gy CONV at 0.5% O_2_ and 20 Gy FLASH at 0.5% O_2_ previously irradiated WB-PBLs (see columns 6 and 8, respectively) with a further dose of 10 Gy CONV under normoxia led to equal measures of an additional Comet response (see columns 7 and 9, respectively), indicating equal levels of further strand break formation. Furthermore, these levels of further strand break formation were not significantly different to each other, nor to the response noted for the single 10 Gy CONV irradiation under normoxia (column 4). The equivalent levels of further strand break damage being induced following the initial FLASH and CONV irradiations indicate no effect of further crosslinks being formed as a result of FLASH (or CONV) irradiation. Consequently, this study failed to provide clear evidence to support the RRR mechanism occurring under FLASH exposure.

Treatment of WB-PBL samples with cisplatin (10–1000 µM), a well-established chemotherapeutic DNA crosslinking agent, led to clear decreases in subsequent radiation-induced Comet formation (Figure 1B). A statistically significant difference (*p* < 0.01) was observed between mean % Tail DNA values of the untreated 10 Gy-irradiated WB-PBL (column 2, Figure 1B) and the WB-PBL treated with the highest concentration (1000 µM) of cisplatin prior to irradiation (column 5, Figure 1B). Similarly, the calculated ‘Percentage Decrease in % Tail DNA (%TD)’ (see Section 4) was indicative of cisplatin-induced crosslink formation (Figure 1C), observed by an increase in the Percentage Decrease in % Tail DNA as the cisplatin concentration increased.

### 2.2. Comet Assay Assessment of FLASH-Induced Base vs. Strand Break Damage as a Possible Marker of Transient Oxygen Depletion

A further variant of the Comet assay, incorporating the use of the oxidatively damaged purine base lesion (ODPBL) recognising endonuclease formamidopyrimidine DNA glycosylase (FPG) [30,35], was used to assess FLASH-induced base vs. strand break damage as a possible indicator of FLASH inducing damage similar to that seen in anoxic samples, a likely outcome of TOD. Following irradiation and overnight lysis (see Section 4), the agarose gels were treated with either enzyme reaction buffer (ERB) alone or with FPG to assess the level of induced strand breaks or ODPBL, respectively (Figure 2). The Comet responses (% Tail DNA) following 20 Gy CONV irradiation at 21, 0.7, 0.5, and 0.35% O_2_ are shown in Figure 2A, with the noted responses for the FPG-treated samples (columns 2, 4, 6, and 8) being consistently greater than the corresponding ERB-only treated slides (columns 1, 3, 5, and 7). This reflects the action of FPG at its ‘sensitive sites’ (FPGss), causing further strand breakage, with these additional strand breakages reflecting the presence of induced ODPBL. For Figure 2A and Figure 3A, the presented data have had the corresponding unirradiated control Comet response subtracted to better portray the actual additional strand breakage caused by the FPG treatment, reflecting the presence of induced ODPBL.

The determined relative levels of induced strand break damage vs. FPGss following 20 Gy CONV irradiation at 21, 0.7, 0.5, and 0.35% O_2_ are shown in Figure 2B. The values for the relative levels of induced strand breaks were determined from the ERB-only Comet response (Figure 2A: columns 1, 3, 5, and 7), whilst the values for the relative levels of induced FPGss were determined by subtracting the ERB-only Comet responses from the corresponding FPG Comet responses (Figure 2A: columns 2, 4, 6, and 8). Clearly, as the oxygen tension was reduced, the relative levels of induced strand break damage were reduced, whilst the corresponding relative levels of induced FPGss increased. Consequently, should FLASH irradiation induce TOD at low oxygen tensions (at ca. 0.5% O_2_), yielding a circumstance of transient hypoxia, the resulting FLASH-induced damage should have a more anoxic profile, with this being reflected as a reduced relative level of induced strand break damage, together with an increased relative level of induced FPGss formation following FLASH exposure.

The Comet responses (% Tail DNA) following 20 Gy CONV and FLASH irradiations at both 21% and 0.5% O_2_ are shown in Figure 3A. The Comet responses noted for the FPG-treated samples (columns 2, 4, 6, and 8) were again consistently greater than for the corresponding ERB-only treated samples (columns 1, 3, 5, and 7), reflecting the presence of induced ODPBL. Figure 3B shows the corresponding relative levels of induced strand breaks and FPGss following the CONV and FLASH irradiations at 21% and 0.5% O_2_. As expected, CONV and FLASH irradiations under normoxia (columns 1 and 2) induced near-equal relative levels of strand breaks and FPGss for both irradiations. This was due to the considerable excess of oxygen present under normoxia (21% O_2_: ca. 280 µM) compared to the maximum likely yield of radiation-induced oxygen-consuming radicals induced by 20 Gy (ca. 15 µM; ca. 0.75 µM Gy^−1^) [36]. However, a comparison of the CONV and FLASH irradiations at 0.5% O_2_ (columns 3 and 4) revealed a much-reduced relative level of induced strand break damage, together with a relative increase in FPGss formation following FLASH irradiation, indicative of a more anoxic profile of damage being induced (see Figure 2B).

### 2.3. BSO Pre-Treatment to Assess the Effect of Lowering Thiol Levels on the FLASH-Mediated Reduced Damage Burden

WB-PBLs were treated with 5 mM of buthionine sulfoximine (BSO) prior to irradiation to assess whether lowering cellular thiol levels was effective in abrogating the reduced damage burden mediated by FLASH exposures at low oxygen tension (Figure 4). As noted above (see Figure 1) and as reported previously [26], 20 Gy CONV irradiation at 0.5% O_2_ (column 5) led to a significantly lower level of induced DNA damage compared to irradiation under normoxia (column 4) (*p* = 0.01), and this was further significantly reduced following 20 Gy FLASH irradiation at 0.5% O_2_ (column 7) (*p* = 0.03). However, treatment of WB-PBLs with BSO prior to FLASH irradiation at 0.5% O_2_ led to a significant increase in DNA damage of 4.6% Tail DNA (column 8) compared to the corresponding non-BSO-treated FLASH irradiated samples (column 7) (*p* < 0.001), which represents a >75% abrogation of the FLASH-mediated reduced damage burden (denoted by the difference between columns 5 and 7). Conversely, BSO treatment prior to CONV irradiation at 0.5% O_2_ led to only a non-significant increase in DNA damage of 3.7% Tail DNA (column 6) compared to the corresponding non-BSO-treated CONV irradiated samples (column 5), representing just a <30% abrogation of the reduced damage that occurred upon lowering the oxygen tension from 21% (column 4) to 0.5% (column 5). The greater relative effect of BSO treatment in abrogating the FLASH-mediated reduced strand break damage burden supports a role for thiols in facilitating the FLASH sparing effect. This, in turn, supports FLASH exposures inducing further hypoxia, which further supports TOD as contributing to the FLASH-induced reduction in damage burden. However, there was still a significant (*p* = 0.03) difference in the levels of induced damage between FLASH (column 8) and CONV (column 6) irradiated samples at 0.5% O_2_, following treatment with BSO, indicating that thiols are not the only factor contributing to the reduced damage by FLASH.

## 3. Discussion

To date, two key radiochemical hypotheses have been proposed to explain the normal tissue-sparing effects of FLASH at the biological level [17,18,20,24]: RRR and TOD.

The RRR hypothesis proposes that the higher concentrations of radicals generated during a FLASH exposure favour their recombination, ultimately reducing the number of radicals available to mediate normal tissue damage. Reports investigating ‘pure’ water systems appear to support the notion that such recombination could involve the products of water radiolysis, due to a greater occurrence of inter-track reactions occurring under FLASH conditions [37,38,39]. However, the doses presently studied are likely not high enough for track overlap to be a factor, and furthermore, the high scavenging capacity of the cellular milieu will undoubtedly reduce the likelihood of any inter-track reactions occurring.

With it being unlikely that RRR is mediated via the primary species of water radiolysis, the role of the subsequently induced, and much longer-lived, secondary and tertiary bio/organic free radical species should be considered. This was highlighted by Spitz et al. [40], who further proposed the importance of Fenton chemistry and peroxidation chain reactions in the differential fate of tumour and normal cells exposed to FLASH dose rates. Building on this, Labarbe et al. [21] proposed a physicochemical model of reaction kinetics which supports peroxyl radical recombination as the main determinant of the FLASH effect. Simulations based on this model also suggest that FLASH irradiation increases the recombination of carbon-centred alkyl radicals (**R•**) (see Reaction (1)), making them less prone to react with the cellular oxygen (Reaction (2)), therefore reducing cellular exposure to peroxyl radicals (**ROO**•). These molecules are known to be a major source of DNA and lipid damage in the cell, so reducing the exposure to these species could be related to the sparing effect of FLASH. Recently, FLASH irradiation has been observed to produce no lipid peroxidation in lipid micelles and liposomes (with pulses greater than 0.2 Gy per pulse for a dose of 40 Gy), as opposed to CONV irradiation which initiated lipid peroxidation. A reduction or absence of lipid peroxidation in FLASH further supports the RRR hypothesis [41].
**R• + O_2_ → ROO•**(2)

Still, a possible expectation of the RRR mechanism is the potential formation of crosslink damage (**R–R**), particularly if organic radicals (R•) have the opportunity to recombine, and should this occur at the DNA, then inter-strand crosslinks may be formed [21]. However, following FLASH irradiation, we see no evidence of any additional DNA crosslink formation (Figure 1A), despite there being clear evidence of crosslink formation following treatment of WB-PBL samples with cisplatin (Figure 1B,C), a well-established chemotherapeutic DNA crosslinking agent. Furthermore, the version of the Comet assay used is sensitive enough to detect clinical levels of chemotherapeutic crosslinks in vivo [28,29]. Consequently, this study failed to provide tangible evidence to support the RRR mechanism occurring under FLASH exposure.

However, our results may not fully discount the RRR mechanism from occurring as we are unlikely to observe any crosslink formation as a consequence of peroxyl radical recombination, as the resulting tetroxide species formed (Reaction (3)) is highly labile and will quickly break down [25,42]:**ROO• + ROO• → R-O-O-O-O-R**(3)

Considering the possible short lifetime of peroxyl radicals in the cellular milieu, due to their reaction with nearby biomolecules, antioxidants [43,44], and/or superoxide elimination [25,42], together with the geometric proximity, orientation, and/or mobility likely required of a pair of peroxyl radicals for tetroxide formation, intuitively, it is difficult to perceive peroxyl radical recombination occurring to any significant degree following a typical FLASH exposure.

With our failing to provide experimental evidence to support the RRR mechanism occurring following FLASH exposure, attention turned to the alternate mechanism of TOD. Oxygen has been described as a key player in the FLASH effect, with some studies reporting that a FLASH effect is mainly observed at oxygen tensions below 12–33 mmHg (1.6–4.3%) [27] or 3.8 mmHg (0.5%) [26], while others report that the protective effect of FLASH can be significantly reduced by increasing the oxygen concentration [45].

Again, in a similar fashion to the above-mentioned studies of RRR, many studies of the radiolytic consumption of oxygen as a driver of the FLASH effect, and notably its sparing of normal tissues, have investigated models utilizing water [38,39,46,47,48,49]. Such studies are based on the known rapid reactions of aqueous electrons (e_aq_^−^) and H• radicals with dissolved molecular oxygen in pure water (Reactions (4) and (5)) [50]:**e_aq_^−^ + O_2_ → O_2_•^−^**(4)
**H• + O_2_ → HO_2_•**(5)

However, the use of ‘pure’ water as either an experimental or in silico model of the intracellular environment is inappropriate [17,51]. This is because the stated mechanisms by which the products of water radiolysis deplete pure water of oxygen (e_aq_^−^ and H• reacting with oxygen to produce superoxide anions and perhydroxyl radicals via Reactions (4) and (5), respectively) will not occur to any significant extent within cells because of the high concentrations of competing scavengers [17].

A more likely route for the radiolytic consumption of oxygen is via oxygen reacting with the radiation-induced secondary and tertiary organic radicals that will occur on the millisecond-or-greater timescale [52]. Here, the greater initial concentration of FLASH-induced organic radicals causes a depletion of local oxygen that occurs on a timescale much faster than tissue/cell re-oxygenation kinetics. This in turn leads to a circumstance of transient local hypoxia, better enabling the chemical repair of damaging radiation-induced free radicals [34,53,54]. The eventual outcome is that FLASH exposures should yield lower levels of induced damage compared to CONV exposures.

Recent studies measuring oxygen in solutions all indicate that more oxygen is depleted during CONV than FLASH exposures [55,56,57]. While measurements in vivo show a measurable change in oxygen content only during FLASH exposure, the consumption of oxygen per Gy is generally considered as too low to have a biological impact that would (solely) explain the FLASH effect [55,56,57].

A possible expectation of the TOD mechanism, and the consequentially induced hypoxia, would be a more anoxic profile of the resulting FLASH-induced damage. Furthermore, if thiol-mediated chemical repair plays a role in enabling the reduced damage burden mediated by FLASH, then lowering cellular thiol levels should serve to abrogate the reduced damage burden. The present study clearly shows a more anoxic profile of damage being induced following FLASH exposure at 0.5% O_2_, being manifested as a much reduced relative level of induced strand break damage, together with a relative increase in FPGss formation following FLASH irradiation (see Figure 3B). This change in the profile of damage was confirmed as indicative of an anoxic profile of damage via CONV irradiations at reduced oxygen levels (Figure 2B). These relative changes in strand break vs. base damage may be accounted for by a simple model, whereby •OH radicals add to DNA bases to generate DNA base radicals (DNA)• reactive towards oxygen (Reaction (6)), with some of the resulting peroxyl radicals ((DNA)OO•) able to abstract H from a nearby sugar, eventually leading to strand breaks (Reaction (7)) [58]. Consequently, if base peroxyl radicals are more proficient than (DNA)• in abstracting H-atoms from a nearby sugar, leading to a greater extent of strand breakage [59], and if FLASH induces transient hypoxia through TOD, then this may account for the reduced relative level of induced strand break damage, together with the relative increase in base damage noted following FLASH irradiation (Figure 3B).
**(DNA)• + O_2_ → (DNA)OO•**(6)
**(DNA)OO• → (DNA)OOH + sugar damage → → strand breaks**(7)

It is also noteworthy that whilst FLASH irradiation leads to a reduction in strand break damage burden, there was no reduction in the overall damage burden detected following FPG treatment, with similar levels of Comet response being noted following either 20 Gy CONV or FLASH irradiation at 0.5% O_2_. This indicates that TOD serves to drive a change in the profile of damage as well as leading to a reduction in strand break damage.

Furthermore, the present study also clearly showed that BSO treatment of WB-PBLs, to lower cellular thiol levels prior to FLASH irradiation, led to a significant abrogation of the FLASH-mediated reduced strand break damage burden, compared to only a non-significant increase in damage that occurred upon BSO treatment prior to CONV irradiation (Figure 4). The greater relative effect of BSO treatment in abrogating the FLASH-mediated reduced strand break damage burden supports a greater role for thiols in facilitating the FLASH-induced reduced damage burden. This in turn supports FLASH exposures inducing further hypoxia, which further supports TOD as a mechanism contributing to the FLASH-induced reduced damage burden. However, there was still a significant (*p* = 0.03) difference between levels of induced damage in the BSO-treated samples irradiated with FLASH (column 8) and CONV (column 6) at 0.5% O_2_. This suggests that although thiols may play a role in mediating the reduction in strand break damage burden induced by FLASH, other factors also contribute.

## 4. Materials and Methods

### 4.1. Comet Assay Assessment of FLASH-Induced DNA Inter-Strand Crosslinks

The alkaline Comet assay is a well-established technique for measuring DNA strand break damage [35], but can be modified to allow for the sensitive detection of DNA inter-strand crosslinks both in vitro and in vivo [28,29] (see Appendix A). The presence of crosslinks retards the electrophoretic mobility of radiation strand-broken alkaline-denatured cellular DNA, compared to non-crosslinked irradiated controls, and therefore crosslinks can readily be detected by the alkaline Comet assay as a decrease in radiation-induced comet formation, with the extent of retardation being proportional to the level of induced inter-strand crosslinking. Using this version of the assay, we have looked for evidence of further crosslink formation in FLASH irradiated WB-PBLs ex vivo (vs. CONV irradiated WB-PBLs) by re-irradiating with 10 Gy CONV under normoxia prior to Comet analysis, to deliver a fixed level of random strand breaks in the genome. The re-irradiation step was conducted under aerobic conditions in order to maximise the yield of strand break damage to best enable the detection of any possible FLASH-induced crosslinks.

Comet slides of WB-PBLs were prepared, incubated under either normoxia (21% O_2_) or 0.5% O_2_, and then irradiated with either 10 or 20 Gy CONV (0.1 Gy s^−1^) or 20 Gy FLASH (2 kGy s^−1^) 6 MeV electrons, as previously described by using a humidified hypoxia chamber and sealing slides inside air-tight vessels prior to removal from the chamber [26]. Following the initial FLASH or CONV irradiations, selected slides were then re-irradiated with 10 Gy CONV under normoxia. After irradiation treatment(s), all slides were placed into ice-cold lysis buffer in preparation for the Comet analysis: electrophoresis, Comet visualisation, and scoring, as previously described [26]. Representative images obtained from the assay are provided in the Appendix A.

As a positive control for crosslink formation, 20 µL WB samples were treated with 5 µL cisplatin solutions (made up in 0.9% saline solution) for 6 h at room temperature, with the final concentration of cisplatin being either 10, 100, 250, or 1000 µM. After cisplatin treatment, Comet slides were prepared and irradiated on ice with 10 Gy 195 Kv X-rays using a RS320 Irradiation cabinet (Xstrahl, Walsall, UK) fitted with a 0.5 mm Cu filter at 0.026 Gy s^−1^. Following irradiation, the standard alkaline Comet assay procedure was undertaken. Crosslink formation induced by incubation with cisplatin was assessed by the relative decrease in the extent of Comet formation noted for the drug-treated irradiated samples compared to the non-drug-treated irradiated WB-PBL. Crosslinking was expressed as the ‘Percentage Decrease in % Tail DNA (%TD)’ compared with irradiated controls, calculated by the formula: % decrease in %TD = [1 − (%TDdi − %TDcu)/(%TDci − %TDcu)] × 100, where %TDdi is the %TD of the drug-treated irradiated sample, %TDcu is the %TD of the untreated unirradiated control, and %TDci is the %TD of the untreated irradiated control [28].

### 4.2. Comet Assay Assessment of FLASH-Induced Base vs. Strand Break Damage

In addition to detecting DNA strand break damage and inter-strand crosslinks, the Comet assay has also been developed to detect various base lesions present in damaged DNA. This involves the introduction of an additional step to the standard protocol, whereby the post-lysis-generated nucleoids are treated with a base lesion-specific endonuclease enzyme that recognizes and cleaves the DNA at the site of the recognised base lesion(s) (see Appendix A) [35]. Consequently, the endonuclease-recognised lesions are converted into strand breaks which are detected as an ‘additional’ Comet response following endonuclease treatment. An endonuclease commonly used in this version of the Comet assay is FPG (formamidopyrimidine DNA glycosylase), which both detects and cleaves DNA at the major purine oxidation products of 7,8-dihydro-8-oxoguanine (8-oxoguanine), 8-oxoadenine, fapy-guanine, and fapy-adenine, as well as at other altered purines [30,60,61].

Comet slides bearing gel-embedded WB-PBLs were prepared (including their incubation under either normoxia (21%) or under reduced partial pressures of oxygen: 0.7, 0.5, or 0.35%) and irradiated with 20 Gy FLASH (2 kGy s^−1^) or CONV (0.1 Gy s^−1^), as previously described [26]. Following irradiation, all slides were placed into ice-cold lysis buffer and held overnight at 4 °C. After cell lysis, each slide-mounted gel was washed three times with 50 µL of the supplied 1× ERB (enzyme reaction buffer) for 5 min. The gels were then treated with either 50 µL of 1× ERB alone or 0.1 U FPG in 50 µL of 1× ERB, with a coverslip, then applied to spread the ERB or ERB+FPG evenly over the gel. The slides were then incubated at 37 °C in humidified light-tight boxes for 30 min. Subsequent electrophoresis, Comet visualisation, and scoring were all undertaken as previously described [26].

### 4.3. BSO Pre-Treatment of WB-PBLs

WB-PBLs were treated with BSO (buthionine sulfoxamine) prior to irradiation to assess whether reducing cellular thiol levels were effective in abrogating the reduced damage burden mediated by FLASH exposures at low oxygen tension.

Aliquots (14 µL) of whole blood containing Na_2_EDTA (to a final concentration of 1.6 mg EDTA/mL of blood) were mixed with 5 µL of 19 mM BSO prepared in a 0.9% NaCl solution, generating a treatment concentration of 5 mM of BSO, and incubated for 4 h at room temperature. Comet slides were prepared, incubated under either normoxia (21% O_2_) or 0.5 % O_2_, and then irradiated with either 20 Gy CONV (0.1 Gy s^−1^) or 20 Gy FLASH (2 kGy s^−1^) 6 MeV electrons, as previously described [26]. Following irradiation, all slides were placed into ice-cold lysis buffer overnight and electrophoresis, Comet visualisation, and scoring were all undertaken as previously described [26].

### 4.4. Statistical Analysis

Statistical analysis was performed using the ‘Analyse’ tool in GraphPad Prism 9 (GraphPad Software, La Jolla, CA, USA). A two-tailed unpaired *t*-test was used to assess statistically significant differences (*p* < 0.05) between mean % Tail DNA values.

## 5. Conclusions

In summary, the current study provided no experimental evidence to support the RRR mechanism contributing to the reduced damage burden induced by FLASH. However, the observation of a greater anoxic profile of damage following FLASH irradiation, together with the BSO abrogation of the reduced strand break damage burden mediated by FLASH, lends further support to the TOD mechanism being involved in the reduced damage burden mediated by FLASH.

## Figures and Tables

**Figure 1 ijms-24-07195-f001:**
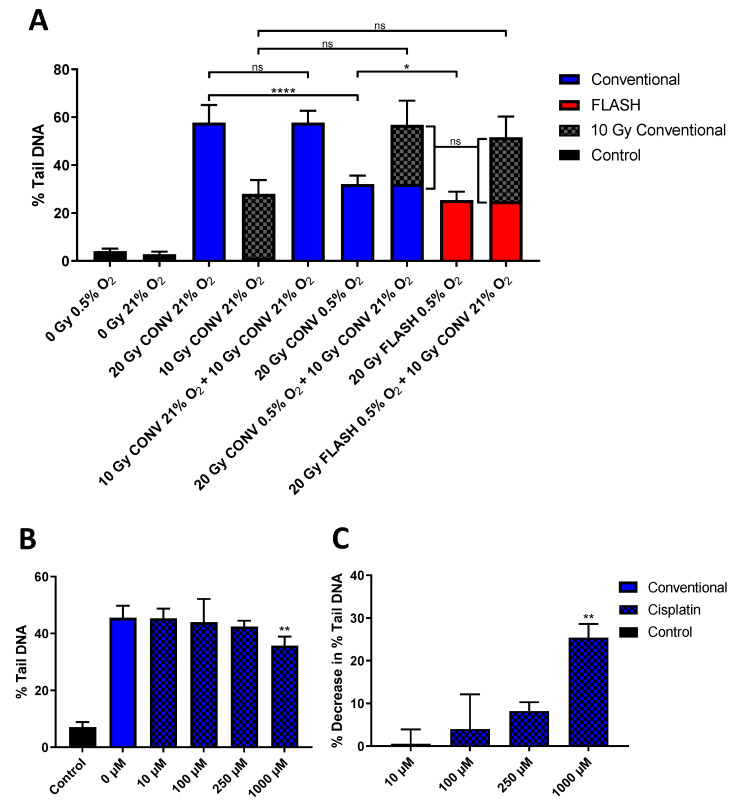
(**A**) Alkaline Comet assay measures of DNA damage formation (% Tail DNA) in whole-blood peripheral blood lymphocytes (WB-PBL) following either 20, 10, or 2 × 10 Gy conventional dose rate (CONV, 0.1 Gy/s) irradiation or 20 Gy FLASH (2 kGy/s), at either 0.5 or 21% oxygen tension, with select CONV and FLASH 0.5% O_2_-irradiated samples, then re-irradiated with 10 Gy CONV under normoxia. (**B**) Alkaline Comet assay measures of DNA damage formation (% Tail DNA) following treatment of WB-PBL samples with cisplatin (10–1000 µM) or control vehicle (0.9% saline solution) for 6 h at room temperature prior to 10 Gy 195 Kv X-ray irradiation at 21% O_2_. (**C**) The calculated ‘Percentage Decrease in % Tail DNA (%TD)’ (see Section 4), being indicative of cisplatin-induced crosslink formation. All data are expressed as the mean % Tail DNA of three slides (*n* = 3). Error bars indicate standard deviation of the means for each experimental condition. Statistical significance (two-tailed unpaired *t*-test) is indicated with *, where * is *p* < 0.05, ** is *p* < 0.01 and **** is *p* < 0.0001, non-significant (ns) is *p* > 0.05.

**Figure 2 ijms-24-07195-f002:**
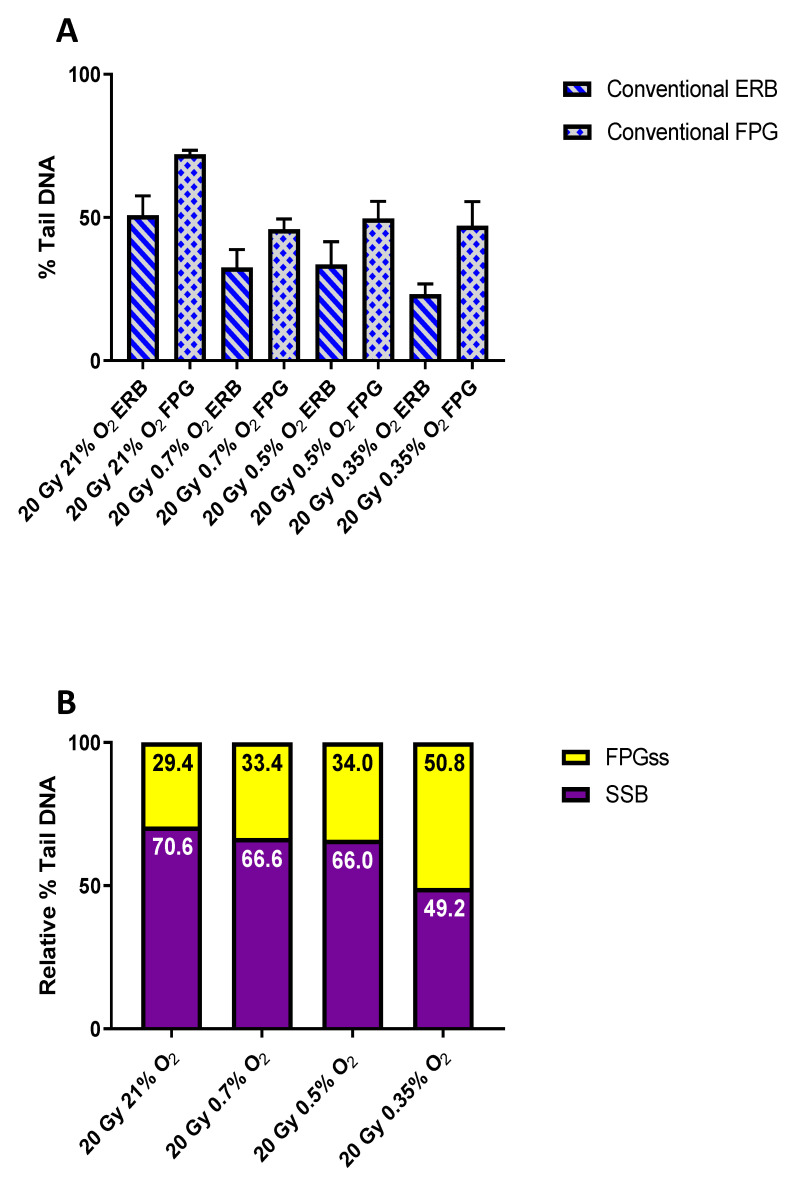
(**A**) Alkaline Comet assay measures of DNA damage formation (% Tail DNA) in whole-blood peripheral blood lymphocytes (WB-PBL) following 20 Gy CONV irradiation at 21, 0.7, 0.5, or 0.35% O_2_ and treatment of the post-lysis-generated nucleoids with either FPG (0.1 U per gel) (columns 2, 4, 6, and 8) or ERB-only (columns 1, 3, 5, and 7). Data are expressed as the mean % Tail DNA of three slides (*n* = 3). Error bars indicate standard deviation of the means for each experimental condition. (**B**) The determined relative levels of induced strand break damage (SSB) vs. FPGss following 20 Gy CONV irradiation at 21, 0.7, 0.5, or 0.35% O_2_.

**Figure 3 ijms-24-07195-f003:**
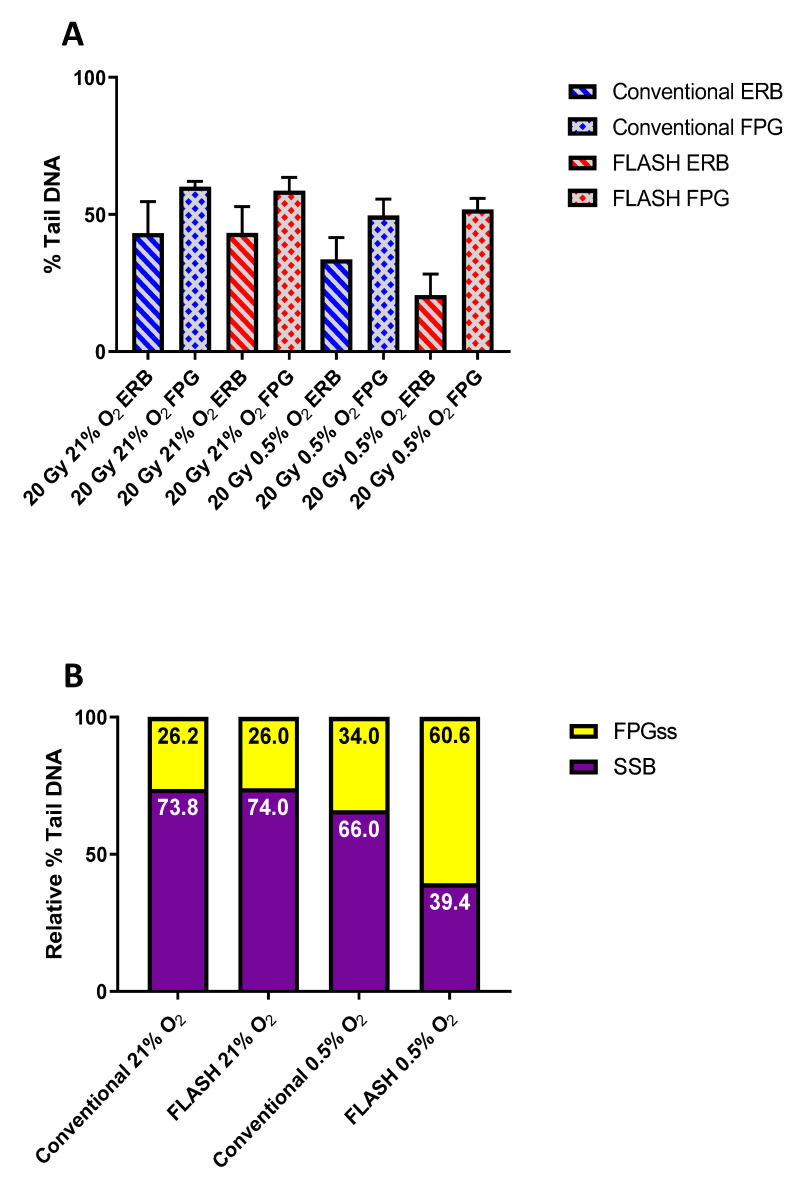
(**A**) Alkaline Comet assay measures of DNA damage formation (% Tail DNA) in whole-blood peripheral blood lymphocytes (WB-PBL) following 20 Gy CONV or FLASH irradiation at 21 or 0.5% O_2_, and treatment of the post-lysis-generated nucleoids with either FPG (0.1 U per gel) (columns 2, 4, 6, and 8) or ERB-only (columns 1, 3, 5, and 7). Data are expressed as the mean % Tail DNA of three slides (*n* = 3). Error bars indicate standard deviation of the means for each experimental condition. (**B**) The determined relative levels of induced strand break damage (SSB) vs. FPGss following 20 Gy CONV or FLASH irradiation at 21 or 0.5% O_2_.

**Figure 4 ijms-24-07195-f004:**
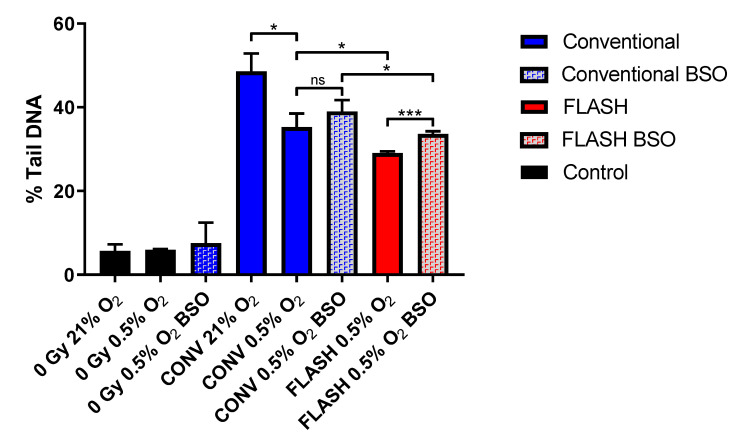
Alkaline Comet assay measures of DNA damage formation (% Tail DNA) in whole-blood peripheral blood lymphocytes (WB-PBL) following 20 Gy CONV or FLASH irradiation at 21 or 0.5% O_2_, after treatment of WB-PBL with buthionine sulfoximine (BSO). Data are expressed as the mean % Tail DNA of three slides (*n* = 3). Error bars indicate standard deviation of the means for each experimental condition. Statistical significance (two-tailed unpaired *t*-test) is indicated with *, where * is *p* < 0.05 and *** is *p* < 0.001, non-significant (ns) is *p* > 0.05.

## Data Availability

Please contact the corresponding authors for any additional information.

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
