# Peer review of "Comet Assay Profiling of FLASH-Induced Damage: Mechanistic Insights into the Effects of FLASH Irradiation"

_ijms, 2023, doi:10.3390/ijms24087195_

Round 1
Reviewer 1 Report
The authors describe in their manuscript in vitro comet assay experiments that were conducted in order to find indications for potential mechanisms of the Flash effect. This study is a continuation of a previous work of the same authors using the same model and assay to detect an electron flash effect and to reveal potential dependencies on dose and dose rate. Dinstict from their previous work the authors apply in the present study two different adaptations of the comet assay to see if radical-radical-recombination or transient oxygen depletion contribute to the reduced normal tissue damage after flash irradiation.
In general, the application of in vitro assays for Flash effect studies is questionable, since Flash is per defintion a differential effect in normal and tumor tissue. However, the authors perform their experiments at reduced oxygen levels, which might reflect the tissue situation and might hint some effects, that needs to be validated in vivo later on.
RRR analysis:
- What is the relevance of DNA-crosslinks compared to other DNA damages like DSB, SSB and DNA-protein crosslinks? Can you provide some numbers to justify the endpoint and method?
- Would the comet assay be able to detect protein-DNA crosslinks?
- I did not understand, why the authors change the oxygen level for the re-irradiation describes to Figure 1. One would not assume such a dramatic change in pO2 within the irradiated tissue.
- Comparing the treatments in Figure 1a and 1B, I wonder if the radiation alone might nor harsh enough to induce a detectable number of DNA-crosslinks. In Figure 1B just the highest dose of cisplatin result in a significant reduction of %tail- DNA. This is correlated to the question above - what is the probability to induce a DNA-crosslink by RRR in relation to other RRR-products.
- In the discussion I clearly miss the comparison to other publications, where RRR is indicated. Exemplarily, Froidevaux, Pascal, et al. ("FLASH irradiation does not induce lipid peroxidation in lipids micelles and liposomes." Radiation Physics and Chemistry 205 (2023): 110733.) have shown that lipid peroxidation was not induced after flash, which yould probably be the result of reduced radical levels through RRR. Moreover, Jansen et al ("Changes in Radical Levels as a Cause for the FLASH effect: Impact of beam structure parameters at ultra-high dose rates on oxygen depletion in water." Radiotherapy and Oncology 175 (2022): 193-196.) provide evidence for RRR in electron Flash treatment.
TOD analysis:
- Unfortunately, Figure 3 is described in the text but not included into the manuscript reviewer version.
- Again, the authors miss to reference important in vivo and in situ data already obtained in the context of TOD analysis. From phantom measurments (e.g. Jansen et al. 2022 see above) to in vivo data (see list below) several groups have shown that less oxygen is depleted during Flash irradiation compared to conventional reference treatment. As such, TOD might have a small contribution to the Flash effect, but is not the major driver. In order to justify the present data, the authors should refer to already existing data that level the contribution of TOD.
Examples for TOD:
- Van Slyke, Alexander L., et al. "Oxygen monitoring in model solutions and in vivo in mice during proton irradiation at conventional and flash dose rates." Radiation Research 198.2 (2022): 181-189
- Ha, Byunghang, et al. "Real-time optical oximetry during FLASH radiotherapy using a phosphorescent nanoprobe." Radiotherapy and Oncology 176 (2022): 239-243.
-El Khatib, Mirna, et al. "Ultrafast tracking of oxygen dynamics during proton FLASH." International Journal of Radiation Oncology* Biology* Physics 113.3 (2022): 624-634.
-Leavitt, Ron J., et al. "Hypoxic tumors are sensitive to FLASH radiotherapy." bioRxiv (2022): 2022-11.
These are just the measured, which are supported and contradicted by numerous simulations of which some were mentioned in the discussion.
General comment:
- The Introduction and the Discussion included repeated text. The same appear in the results section, where most of the text in the captions is also given in the text. This makes the manuscript unnecessarily long and hard to read. All this numbers given in the results are depicted in the figure. Maybe a clever table would make it better readable.
Reviewer 2 Report
The authors probably wrote their work with M&M inserted just after Introduction and abbreviations are explained in M&M. But now abbreviateions must be explained at the initial appearance, please modify it accordingly. In Figure 1-B and 1-C captions must also include that X-irradiation was delivered after CDDP treatment.
Round 2
Reviewer 1 Report
The authors did a great job to revise the manuscript and answering the question in an understandable way. They should think about inclusion of some of the information from the response letter in the manuscript; not everyone wants to read all references to understand that the re-irradiation is part of the assay (e.g.).
Anyway, the new diagrams in the supplement nicely describe the comet methods applied in the study and illustrate the outcome.
The discussion clearly benefit from the more balanced discussion that better reflect the ongoing research in the Flash field. The removal of doubled information in the figure caption also increase the readability.
Minor points:
l49-55 Introduction: The short summary on the in vivo experiments and the clinical trial is nice, but references are missing.
l321-324 Discussion: Again, no references in the reviewer version.
Author Response
Thanks for your feedback, please see the attachment

Reviewer 3 Report
Authors have satisfactorily answered all the questions.
